

# Crossing the Chasm: How to develop weather and climate models for next generation computers?

Bryan N. Lawrence[1,2,3], Mike Rezny[4], Reinhard Budich[5], Peter Bauer[6], Jörg Behrens[7], Mick Carter[8], Willem Deconinck[5], Rupert Ford[9], Christopher Maynard[8], Steven Mullerworth[8], Carlos Osuna[10], Andrew Porter[9], Kim Serradell[11], Sophie Valcke[12], Nils Wedi[6], and Simon Wilson[1,2,8]

[1]Department of Meteorology, University of Reading, UK.
[2]National Centre of Atmospheric Science, UK.
[3]STFC Rutherford Appleton Laboratory, UK.
[4]Monash University, Australia
[5]Max Planck Institute for Meteorology, Germany
[6]ECMWF, UK
[7]DKRZ, Germany
[8]Met Office, UK
[9]STFC Hartree Centre, Daresbury Laboratory, UK
[10]ETH, Switzerland
[11]Barcelona Supercomputing Center, Spain
[12]Centre Européen de Recherche et de Formation Avancée en Calcul Scientifique, France

*Correspondence to:* Bryan Lawrence (bryan.lawrence@ncas.ac.uk)

**Abstract.**

Weather and climate models are complex pieces of software which include many individual components, each of which is evolving under the pressure to exploit advances in computing to enhance some combination of a range of possible improvements (higher spatio/temporal resolution, increased fidelity in terms of resolved processes, more quantification of uncertainty etc). However, after many years of a relatively stable computing environment with little choice in processing architecture or programming paradigm (basically X86 processors using MPI for parallelism), the existing menu of processor choices includes significant diversity, and more is on the horizon. This computational diversity, coupled with ever increasing software complexity, leads to the very real possibility that weather and climate modelling will arrive at a chasm which will separate scientific aspiration from our ability to develop and/or rapidly adapt codes to the available hardware.

In this paper we review the hardware and software trends which are leading us towards this chasm, before describing current progress in addressing some of the tools which we may be able to use to bridge the chasm. This brief introduction to current tools and plans is followed by a discussion outlining the scientific requirements for quality model codes which have satisfactory performance and portability, while simultaneously supporting productive scientific evolution. We assert that the existing method of incremental model improvements employing small steps which adjust to the changing hardware environment is likely to be inadequate for crossing the chasm between aspiration and hardware at a satisfactory pace, in part because institutions cannot have all the relevant expertise in house. Instead, we outline a methodology based on large community efforts in engineering and standardisation, one which will depend on identifying a taxonomy of key activities — perhaps based on



existing efforts to develop domain specific languages, identify common patterns in weather and climate codes, and develop community approaches to commonly needed tools, libraries etc — and then collaboratively building up those key components. Such a collaborative approach will depend on institutions, projects and individuals adopting new interdependencies and ways of working.

# 1  Introduction

Weather and climate models have become ever more complex as a consequence of three main drivers: more computing capacity, greater understanding of important processes, and an increasing mathematical ability to simulate those processes. The first of these has been based on incremental progress using "more of the same", with extra computing capacity coming from a combination of faster clock speeds and accessible parallelism. This capacity has allowed more science areas or processes

("components") to be assembled together, each of which is supported by an expert community who have built and evolved their codes based on adding parameterisations or increasing the amount of direct numerical simulation (of "resolved" processes).

In general, these components are within, or work with, one or two key dynamical "cores" which solve the Navier-Stokes equations to simulate fluid flow in the domain of interest (generally the atmosphere and/or ocean). A complex software infrastructure exists to "join" the components and exploit the parallelism. Developing, maintaining, and executing these models

is extremely challenging — with leading codes having hundreds of thousands to millions of lines of code (Alexander and Easterbrook, 2015) supported by large teams of software engineers and scientists over very long timescales (Easterbrook and Johns, 2009), often with a lack of useful code management tools (Méndez et al., 2014).

In the years ahead, the "more of the same" approach will break down as trends in computer architectures suggest a plethora of heterogeneous and complex computing platforms will be the method of providing more capacity. The "free lunch" arising from

faster clock speeds and easily accessible parallelism is over: more capacity will require exploiting more complex computer architectures and exploring new routes to parallelism. This will occur even as the science codes themselves become more complex and demanding. It is likely we will see new computing hardware arrive and be deployed in a timescale of (at most) years, while, if we carry on as we are, the fundamentals of our code will be changing on decadal timescales at best. This mismatch of timescales can be thought of as an impending cliff: what if we simply cannot migrate our weather and climate

codes in the interval between finding out about a new computer architecture, and its retirement?

In this paper we assert that the challenge ahead can be thought of as analogous to the problem of bridging a vast chasm, where neither a single leap nor a set of small safe steps are practicable. The chasm exists between our science aspirations and the underlying hardware. Our existing approach of incremental advances may be a dead end given that no one group has the resources to take a single giant leap. However, we believe the problem is tractable, and the way forward involves a better

separation of concerns between science, infrastructure, and platform dependent code, the development of a set of well defined tools in each area, and clear and appropriate interfaces/use-cases for the tools. With such a separation of concerns, the problem of crossing the chasm is reduced to a series of (coding) spans between stable platforms (tools and libraries). Rapid evolution in any one area (science code or computer architecture) is isolated from other parts of the system.



In this paper we look at the implications of some of the different sources of complexity ahead and some of the areas where stable platforms might be possible. In doing so we summarise some of the recent progress in relevant tools and components. We conclude that in the final analysis, like getting across any great chasm, the solutions which will need to underpin our future models will involve large community efforts in engineering and standardisation; they won't be built and/or sustained by

relatively small groups acting alone — especially groups with the prime purpose of advancing science as opposed to software.

## 2   Complexity

There are scientific and societal drivers to continuing to improve the fidelity of weather and climate simulations. These increases in fidelity are achieved by adding more important processes, but also by increasing ensemble sizes, and improving the quality of the simulation of existing processes. The latter is accomplished by using higher resolution, better numerics and better quality

initial and/or boundary conditions (that is, making better use of observational data, including the use of data assimilation). In general, better numerics leads to increasing numerical complexity associated with, for example, unstructured and extruded meshes, multiple meshes, multi-grid methods, and higher-order mixed finite element methods. Better initial conditions arise from complex mathematical assimilation techniques. (Larger ensemble sizes and higher resolution lead to large volumes of data — indeed it's clear that handling exabytes of data will be a problem for the community before it has to deal with computers

capable of exaflops — but data handling is not the topic of this paper.) Many of these axes of improvement involve "scientific business as usual", but in the remainder of this section we address areas where ongoing incremental improvement is unlikely to be productive.

Firstly, these scientific advances need to exploit the changing computing environment, where we also see increasing complexity: more cores per socket, multiple threads per core, the changing nature of a core itself, complicated memory hierarchies,

and more sockets per platform. In the future any given platform may assemble compute, memory, and storage in a unique assemblage of components, leading to major code development and maintenance problems associated with the increasing need for engineering resources to exploit potential performance, particularly in energy efficient, robust, and fault-tolerant ways. We review these issues and some of the consequences for weather and climate modelling in section 2.1 ("hardware complexity") below.

Given this background, numerical teams have been looking at computing trends, and making algorithmic choices aimed at efficient execution on future platforms. Amongst such choices are horizontally explicit methods (which are more likely to exploit only neighbour communications) and horizontally semi-implicit or implicit schemes (often with solvers requiring more extended communication, potentially across all processors). The balance of compute time (potentially more in explicit methods) versus communication time (potentially more in implicit methods), and hence overall execution time, will vary

depending on the algorithmic implementations and how they interact with the capability of the hardware. Hence, the best mathematical methods to use at any given time are going to be slaves to the nature of the hardware and the implementation quality — which will be problematic if the hardware is changing quicker than the mathematics and its implementation. We do



not further consider the mathematics here, beyond highlighting the importance of facilitating the ability to quickly translate improvements in mathematics to efficient usable code — one specific application of the separation of concerns we advocate.

One aspect of such usability will be the ability for the mathematics to be transposed into code which addresses concurrency at many more levels than we have done in the past, and this is happening while we are adding more and more physical processes
and model components. We discuss this aspect of the problem in section 2.2 ("Software Complexity") by introducing exemplars of component and software complexity and how they are handled in current models, before briefly summarising the impact of process complexity on the need for parallelism in section 2.3 ("Where is the concurrency?").

## 2.1 Hardware Complexity

Significantly, after many years of "computational similarity" based on "multi-core" Intel or Intel-like X86 systems, the future
looks far more heterogenous than ever. There are a plethora of new architectures available, from evolutions of X86 with more complex vector processing units such as the Intel Many Integrated Core (MIC, or "many core") nodes, to new ARM server chips and/or radically different architectures based on Graphical Processing Units (GPUs). More exotic processor families are emerging too: from Field Programmable Gate Arrays (FPGAs) to new technologies such as Google's Tensor Processing unit (Jouppi et al., 2017). At the time of writing, one vendor alone, Cray, is offering very different nodes in their HPC line:
multi-core CPU alone, many core nodes, or nodes which combine CPUs with GPUs. The community is facing a "Cambrian explosion" in computer architectures.

This proliferation of architectures is a direct consequence of the limitations of physics and power constraints in silicon: substantial year-on-year increases in clock speed are no longer possible, and increased compute capacity has to come from innovation in what is on the chip rather than how fast it goes; more hardware proliferation is expected! An additional driver is
that the technology choices which arise from this proliferation are driven by economics, not the needs of the HPC community. Further, for big machines, power constraints dominate thinking, with existing CPU architectures consuming too much power. The US exascale programme has set a goal of 20MW for an exaflop machine, but this would not be achievable with current technology (Kogge, 2015), leading to more pressure on hardware change — change which is likely to require lower power processors and more of them.

With more cores per chip comes a requirement for more memory, which cannot be met by current technology trends, particularly memory with low latency and high bandwidth. The hardware solutions being offered include a range of types of memory each with their own characteristics, requiring new techniques to manage and control memory interference between different cores, agents, and applications that share the memory system (Mutlu, 2015). Whether caching algorithms will provide the appropriate performance, or programmers will need tools to directly manipulate data in these systems is not known! Further,
this "near memory" is now being coupled with large fast storage "close" to the computing (so called "burst buffers"), leading to nested hierarchies in a spectrum from "memory" to "archive", and new problems (and opportunities) for the workflow around applications.

In most modelling groups, the "modeller" currently has to have full knowledge of how the entire software stack works with the hardware even if implementation details are buried in libraries. The new architectures present the "full-stack" modeller



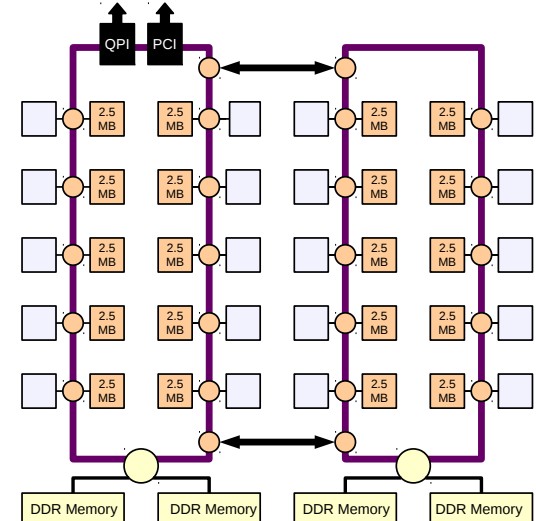

a) Schematic of a single socket Intel XEON 20 core Broadwell processor (typically two of these to a CPU node, mixed nodes may have one of these and an accelerator in another socket).

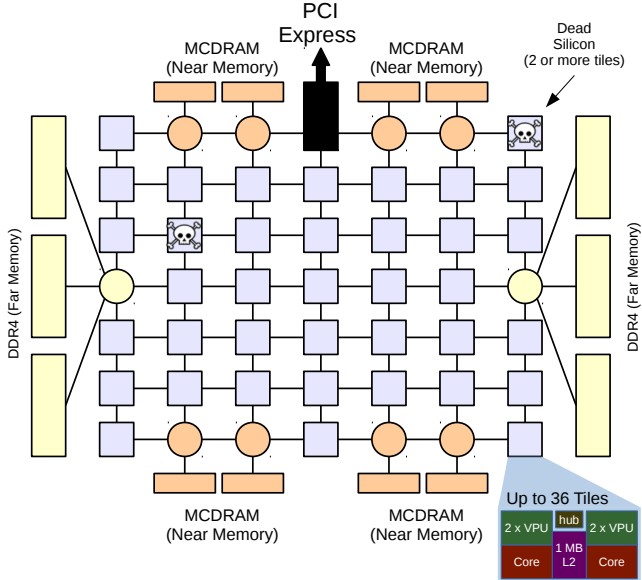

b) Schematic of Intel MIC Knights Landing processor node with 38 tiles (36 usable), each containing two cores each with two vector processing units, and sharing a 1 MB L2 cache.

**Figure 1.** Architectural heterogeneity: A schematic comparison between an Intel Broadwell CPU socket (two sockets to a node, joined locally by the quick path interconnect, QPI) and an Intel Knights Landing MIC node. Notable differences include the two-dimensional mesh linking cores and memory (near and far) on the MIC node (with up to 16 GB of near memory L3 cache), and the bidirectional full ring on the CPU linking the cores (and supporting up to 50 MB shared L3 cache). The most important difference however is the number of cores o(20) to a socket or 40 to a CPU node, and up to 72 on the MIC node. (Not shown on this diagram is the 256 MB L2 cache in the Broadwell cores and their internal vector processing units.)

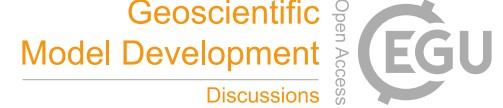



many challenges: Can one code to exploit the parallelism and the memory? Is there sufficient parallelism in the code itself to exploit the architecture? How can one obtain performance data upon which to make informed choices? Can one code to exploit multiple different architectures (and maintain that code)?

Figure 1 shows just one comparison of two relatively similar architectures, a multi core evolution of the traditional Intel

CPU processor, and a many core node. This schematic is intended to show both the similarities and differences: both consist of multiple cores, but there are different arrangements of cache and main memory, different ways of accessing vector processing units, and different methods for linking between the cores.

Today, at the most fundamental level, modellers rely on the compiler to hide most of these details, but with the advent of coding for GPUs (of which more below), it has become clear that the better one can express parallelism in the high level (e.g.

Fortran) code, the better job the compiler can do even on traditional CPUs. Traditionally (say ten years ago), in a weather or climate model one had two levels of concurrency in mind associated with shared and distributed memory:

– Shared memory concurrency involves organising the code so the compiler could be pre-fetching memory while doing calculations, i.e. worrying about loop order etc, and organising loops, array declarations etc to enable the efficient use of vector hardware. This requires "good coding practice", either gleaned by direct knowledge about how compilers and

hardware work, or simply by edict: e.g. "always code nested loops with the innermost loop using the left-hand index[1], and never mind why".

– Distributed memory concurrency typically requires explicit handling in the science code. Processes (typically on different nodes) cooperate on a common task utilising memory which is partitioned across nodes ("domain decomposition") and exchange updates using the Message Passing Interface (MPI, Gropp et al., 1996, ideally using libraries that vendors

have optimised for their interconnects).

With the advent of the multi-core CPU (moving from dual core, to quad core, to now, 20+core) the modeller had to become aware of strategies to exploit the fact that the processor local cores shared memory and were connected together with lower latency and higher bandwidth than the interconnect between processor sockets and nodes.

In most codes now we see one of two strategies: either MPI is still used across both nodes and node-local cores, or MPI is

used in conjunction with OpenMP (Dagum and Menon, 1998). In the latter case OpenMP is used to run *some* parallel threads on a core ("hyper threading") or parallel threads on a socket, but the threads are sharing memory, and directives are used to identify where this can occur. However this strategy of mixing explicit library calls (MPI) and compiler directives (OpenMP) does not always work better than MPI alone, not least because vendor implementations of MPI can implement on-node "messages" with highly optimised libraries exploiting shared memory with performance that can be hard to beat. Conversely, MPI decomposition

cannot be changed without paying a synchronisation and data movement penalty, whereas OpenMP can exploit dynamic scheduling of work, and better handle load balancing around (for example) physics parameterisations such as precipitation. Hence, the modeller following a hybrid approach has many issues to consider: how many parallel threads per node (between zero and the number of hyper-threads supportable on the node) and what parts of the code can take advantage of this technique?

---

[1]...and remember that rule is language dependent, in another language it might be "innermost loop using the right-hand index".





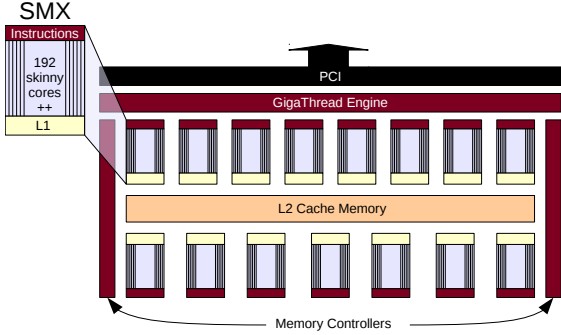

**Figure 2.** High level schematic of an NVIDA Kepler GPU (adapted from NVIDIA, 2012). Up to 15 streaming multiprocessor (SMX) units share a 1.5 MB common dedicated L2 cache, and up to six memory controllers allow access to main memory. Thread control exists across the entire chip (the GigaThread engine) and in each SMX. The SMX units include 192 single precision cores, 64 double precision cores, as well as other special function units, and an internal interconnect so that blocks of 32 threads can be managed internally. They also include 64 KB of L1 memory that can be configured in a combination of shared memory or L1 cache, and 48 KB of read only L1 memory.

There is a balance to be struck which depends on knowledge of the code, the hardware, and the quality of the vendor supplied compilers and libraries — and this balance needs to be re-evaluated for each architecture and model version!

Other programming models, such as the Partitioned Global Address Space (PGAS), try to address the problem of mixed mode programming by a single abstraction to address both shared and distributed memory. Implementations include the C extension, Unified Parallel C; Co-Array Fortran, Chapel, and the (Open)SHMEM library. However, all of these suffer from the lack of adoption and lack of universal availability, both of which contribute to the lack of wide use.

OpenMP also provides task concurrency, where pools of threads are allocated amongst a set of tasks that can be done concurrently. However, in Climate and Weather models, this feature is not widely used.

Now our hypothetical modeller is dealing with at least two levels of concurrency, probably a new directive syntax for declaring what parts of the code can use OpenMP or a completely different programming model (PGAS). She may also be having to deal with multiple languages (e.g. some combination of C++/Fortran and C) to best use the hardware and libraries available.

Most processors have vector units, and the trend is for these to grow both in size (width) and number in new processors. This provides another hardware feature that needs to understood and exploited and further complicates the task of model development. Then of course there are GPUs, which many think of as "big fat vector processing units", but ones where the needed levels of data parallelism are very high, and where out-of-order memory fetches are very expensive. However, a cursory look at figure 2 shows that GPU architectures are themselves complex, with sophisticated memory hierarchies and onboard methods of thread management.

The big difference between the CPU architectures (whether multi-cores or many-cores such as MIC) and the GPUs is the amount of intelligence (complexity providing extra instruction level parallelism and caches to hide memory latency etc) in the cores and on the package around the cores, and how accessible that intelligence is to the programmer (or how easily it can be



exploited by the compiler without exposing hardware concepts to the high level code). In CPUs, much of the silicon is devoted to supporting complexity in the algorithm (branching multiple instructions etc) and in GPUs, much of the silicon is devoted to doing the same thing to lots of data. However, the compiler directives that support this parallelism are still rather different and conform to differing, rapidly evolving, and sometimes poorly defined standards, and so the programmer potentially has

to understand not only both hardware architectures, but how best to exploit them. Requiring all modellers to have detailed hardware knowledge — and keeping such knowledge up to date — is obviously not a tenable proposition!

Compounding all these difficulties, future processors are likely to have to operate at both low power and high processor counts - both factors likely to lead to increased node failure rates. Existing software tools expect perfectly reliable hardware, with checkpointing as the only viable strategy to deal with hardware failure, so there will be need to be new mechanisms for

handling hardware failure, which will undoubtedly add further complexity to the hardware programming environment.

## 2.2 Software Complexity

Weather and climate models traditionally differed primarily in three areas:

1. The duration of a simulation (days versus years),

2. Component complexity (climate models typically include more components associated with longer-time scale pro-
cesses), and

3. How they are initialised (weather models by complex data assimilation systems, climate models by very long duration "spin-up" runs for components with poor observational coverage).

These differences have become blurred as weather prediction has begun to exploit ocean, snow and ice models, and with the advent of seasonal prediction, but for the purposes of the discussion here, what they have in common is more important anyway:

1. The importance of executing the model at a satisfactory speed (e.g. a year/day),

2. Multiple interconnected but often independently developed sub-models (oceans, atmospheres, wave-models etc), sometimes with extra versions to provide nested grids and or local refinement.

3. An interconnection framework or coupler (to exchange information amongst model components),

4. At least one component with an expensive dynamical core (atmosphere and/or ocean), with

5. A significant number of physics routines (radiation, microphysics, clouds, gravity waves etc),

6. An internal diagnostics system (to accumulate variables or calculate extra diagnostics beyond the prognostic variables carried in the model components),

7. Systems to manage ensemble methods (for the extra simulations needed to address uncertainty), and

8. An I/O sub-system (to manage efficient input and output).





These common attributes can be linked together in very complex ways.

At the high level ,Alexander and Easterbrook (2015) showed some permutations of how large components can be coupled together in eight earth system models, and at the same time indicated the relative complexity of the key components. In general, the most complexity was found in the atmospheres of these models, although some earth system models of intermediate complexity (EMICs) were dominated by oceans. The relative distribution of complexity also differed significantly between models aiming at the chemistry thought to be significant in coming decades and those aiming at longer time-scales processes associated with the carbon-cycle.

The interconnection between all the large scale components requires a specialised framework and/or coupler to provide methods for exchanging fields between components. Valcke et al. (2012) provides a comparison of coupling technologies used in CMIP5 – but it is interesting to note that of the six technologies discussed in this paper, only two (ESMF, Theurich et al. 2015, and OASIS, Craig et al. 2017 ) have any widespread use, and that many large institutes feel it necessary to build their own coupling systems.

One key piece of software complexity arises from the a priori necessity to find parallelism within the large components such as the atmosphere and oceans of any but the lowest resolution models. With only a handful of large-scale components, and machines with hundreds to thousands of nodes, exploiting the machine efficiently requires splitting components across nodes exploiting "domain decomposition" — splitting the multi dimensional (potentially all of lon, lat, height) space into "domains" each of which is integrated forward each time step on just one node (with the necessity of passing information between nodes after each timestep so that "cross-domain" differences can be calculated).

The relationship between the dynamics and physics within an atmosphere is also complex, with even more scope for different arrangements of data flow between internal components and processes. Figure 3 provides a glimpse into the interplay of dynamics and physics in three different atmosphere components, from the ECMWF Integrated Forecast System, IFS (IFS 41R1, https://www.ecmwf.int/en/forecasts/documentation-and-support), the GFDL HiRAM (2012 public release, https://www.gfdl.noaa.gov/hiram/, Zhao et al. 2009), and the Met Office's UM (V6.6, Davies et al. 2005). This figure describes the flow of data associated with model prognostics within a model timestep (it does not describe actual code blocks or expose details of the internal time-splitting and it is important to note that any implicit parallelism shown may not be mirrored in the model code).

IFS is a spectral transform model. The key property of such a model is that due to the properties of the Laplacian operator, equations for vorticity, divergence and diffusion can be easily calculated in spectral space, with advection and physics calculated on a grid distributed in physical space. The model exploits two time-levels and a semi-implicit, semi-Lagrangian time-stepping routine. The timestep begins with variables held in spectral space, which are then converted back to grid point space for the physics to be executed sequentially, after which they are converted back to spectral space via a fast Fourier transform (FFT) in longitude and a Legendre transform (LT) in latitude to spectral space, where the Helmholtz equation resulting from the semi-implicit time-stepping is solved and where horizontal diffusion is computed.

HiRAM, the GFDL High Resolution Atmospheric Model is a gridpoint model based on finite volumes and the cubed-sphere grid. The dynamics uses a two-level split explicit time-stepping scheme. The physics components use an implicit algorithm







**Figure 3.** A programmers view of data flow and the interaction of dynamics and physics in the atmosphere of three prominent models: ECMWF's IFS, GFDL's HIRAM, and the Met Office's UM. Red boxes show start and finish of timesteps, yellow boxes show dynamics, blue boxes increment physics, beige boxes physics with full values, and external coupling/library routines in white boxes. Not all prognostic variables and associated routines are shown!

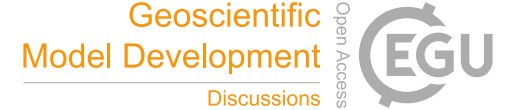



in the vertical (including fast coupling with land and ocean surfaces) integrated using a tridiagonal solver which splits into a down and up sequence. The down, dry processes are calculated first with the increments from the physics and dynamics solver calculated in parallel and then summed. The down physics processes are calculated sequentially. These are then summed and fed into the sea ice and land surface, and then the up, moist processes are calculated; again the increments from the physics

and dynamics solver are calculated in parallel and summed at the end of time-step.

The UM is also a grid point model. In this case the physics is split into "slow" and "fast" processes, which are calculated before and after the main dynamics, which also exploits a semi-implicit semi-Lagrangian time-stepping scheme. The slow physics processes use the state of the previous time-step, but the faster processes are integrated using the latest state calculated sequentially using the incremented versions.

It can be seen that these three examples show rather different methodologies, each with their own take on how to organise the sequence of physical processes. They each involve a vast number of independent pieces of code, each of which could need configuration, compiler or even hardware specific optimisation to take advantage of any given hardware environment.

Although Figure 3 does not show actual internal concurrency, it is clear that there is potential for internal concurrency; the UM for example, does not use the updated state of the variables for the slow physical parameterisations so they could be

calculated in parallel, and some sub-components, e.g. radiation, are expensive enough that they may not be executed every timestep. Such sub-components are large enough to treat, in terms of concurrency, in the same way as full model components —- Balaji et al. (2016) show results from experiments with the same code family as HiRAM, doing exactly that. Their results show that this extra concurrency can be exploited in a range of ways (to shorten the radiation timestep, or to speed up model integrations, albeit with a potential increase in resources used).

Model formulations take a range of approaches to providing diagnostic, ensemble, and I/O services, with no common practice as far as we are aware — beyond some emerging community use of the XIOS I/O server (http://forge.ipsl.jussieu.fr/ioserver/ wiki). I/O servers provide a method of off-loading (primarily) output to one set of processors to happen in parallel with computations happening on other processors. XIOS provides an XML interface which allows not only output offloading, but some diagnostic processing en-route, and work is underway to extend it to support ensemble management as well.

## 25 2.3 Where is the concurrency?

The most obvious paradigm for concurrency in weather and climate models is the domain decomposition introduced above, exploiting MPI, but clearly there are multiple levels of parallelism being provided by the hardware and/or exploitable in the software:

1. Vectorisation within a GPU core or accelerator (using compiler directive languages such as OpenACC), with or without

2. Vectorisation within a CPU core (handled by the compiler, but only if the code is prepared appropriately, possibly by embedding explicit compiler directives),

3. Shared parallelism across CPU and accelerators/GPUs (e.g. see Mittal and Vetter, 2015).

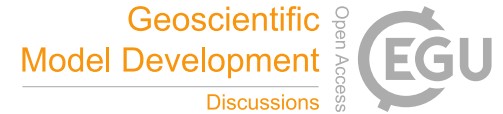

4. Threading providing shared memory concurrency within nodes/sockets (e.g. by inserting OpenMP directives in code),

5. Distributed memory concurrency across nodes, either by

   – utilising MPI (traditional "domain decomposition"; directly, or with a library such as YAXT or GCOM), or

   – exploiting an implementation of a partitioned global address space, such as coarray Fortran (e.g. Mozdzynski et al., 2012, or language neutral interfaces, such as GPI, Grünewald and Simmendinger, 2013).

6. Component concurrency, whether using a framework such as ESMF or handled manually with, for example OpenMP tasks (e.g. Balaji et al., 2016).

7. Coupled model components, either

   – as part of coupled model, executed independently and concurrently using a coupler such as OASIS, or executed together and concurrently using a framework, Valcke et al. 2012) or

   – concurrent models running as part of a single executable ensemble, hand crafted (e.g. Bessières et al., 2016) or via a framework such as the GFDL Flexible Modelling System used in Zhang et al. (2007).

8. I/O parallellism (using an I/O server such as XML IO Server, XIOS, or a parallel I/O library such as parallel NetCDF, Li et al., 2003).

Most of these levels of parallelism require decisions as to where and how the code should be refactored and/or restructured to take advantage of them, and it may not be possible to utilise them all in one code stack, not least because on any given platform implementations may not exist or they may be poorly implemented (in terms of performance or feature coverage).

In the discussion to follow we follow the nomenclature introduced by Balaji et al. to term the first three of these modes as "fine-grained" concurrency, and the last four as "coarse-grained" concurrency. The key distinction between these modes of concurrency is that the former involve organising codes to take advantage of Single Instruction Multiple Data (SIMD) hardware (such as vector units), and the latter involve higher level assemblages interacting — mainly using the Multiple Programme Multiple Data (MPMD) pattern (except for the distributed memory options which utilise the Single Programme Multiple Data, SPMD, paradigm). We do not categorise threading (#4) as either fine or coarse grained concurrency, since in practice from a programmers point of view it can be used following either mode (e.g. to parallelise loops in the same way as one parallelises them for a vector processor or to provide extra models of parallelisation for domain-decomposition).

Current best practice for addressing these modes of concurrency is to

1. Code for hierarchies of parallelism (loops, blocks, domains),

2. Use standard directives (OpenMP/OpenACC),

3. Optimise separately for many-core/GPU, and

4. Try to minimise code differences associated with architectural optimisations.



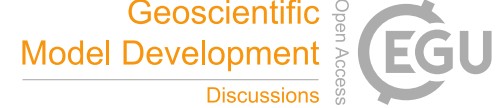

However, this is no longer seen as a long-term successful strategy — at least on its own, not least because the strategies needed for different architectures may be fundamentally different (e.g for memory optimisations). With the advent of exascale systems, entirely new programming models are likely to be necessary (Gropp and Snir, 2013), potentially deploying new tools, or even the same tools (MPI, OpenMP), to deliver entirely new algorithmic constructs such as thread pools and task-based parallelism

(e.g. Perez et al., 2008). Weather and climate models will undoubtedly target such systems and so their disruptive influence is likely to necessitate the wider use of these new programming models.

## 3   Progress

None of the domains of complexity listed above were unexpected: the clock-speed free lunch was over before the end of the first decade of the current millennium, and models have always been evolving by adding processes and more sophisticated

mathematics in their dynamical cores. Since the first vector registers in supercomputers, the pursuit of concurrency has been the key to running fast models, and the transition from massively parallel complex cores to even more massively parallel simpler cores with smaller local memory began with the BlueGene (e.g. Dennis and Tufo, 2008) and has accelerated with GPU entering mainstream compute.

Early work with GPUs began with computationally intensive physics modules where it was clear that different approaches

may be possible in different components, and some of the first investigations into the use of new architectures took this route by rewriting particular components, e.g. Michalakes and Vachharajani (2008) and Hanappe et al. (2011). The latter rewrote (in another language) the radiation code in a widely used fast low resolution climate model for a custom chip - an effort which took 2.5 person-years and delivered a significant speed up on the new processor over the original version. However, although this rewrite achieved significant speed-ups even in the CPU version, it is no longer being actively used — primarily because of the

downstream technical and scientific burden associated with assimilating and maintaining (both technically and scientifically) a rewrite in another language. This downstream burden is one of the main issues associated with re-engineering code — if the code "owner" is not engaged, then rewrites are unlikely to survive.

By contrast, Fuhrer et al. (2014b), in a partnership between meteorologist code owners and computer scientists, have rewritten ab initio the dynamical core of the COSMO weather forecasting model using a domain specific language approach (see

section 3.1) — and it is now operational on a hybrid CPU/GPU machine. Their team is now working on extensions into the physics. One of the lessons learned was that, as with the Hanappe et al. experience, rewriting for good GPU performance resulted in speedups on the CPU as well.

Govett et al. (2014, 2017) have compared and contrasted several generations of compiler implementations of OpenACC and OpenMP on a range of hardware using their next generation Non-hydrostatic Icosahedral Model (NIM). While they have suc-

cessfully implemented a single-source code model which supports a range of hardware platforms, their experiences exposed multiple compiler issues, and depended on ab initio design constraints on their Fortran source code (e.g. basic language constructs and minimal branching). Like Fuhrer et al. their initial work has concentrated on the dynamical core, and one suspects this code cannot be easy to maintain — examples show that in places the number of directives in a code block exceed the



number of lines of Fortran. This would be consistent with the observation that wide disparities in coding effort between the various directive based approaches exist (e.g see Memeti et al., 2017, for a comparison of coding effort in terms of lines of code). The NIM work on radiation codes yielded less than compelling results (Michalakes et al., 2016) as memory bottlenecks were identified, demonstrating that more work was needed on invasive and time consuming refactoring to ready codes for

emerging hardware. Experience in other groups also suggests that unless this work is carried out by parallelisation experts, gains in performance may be outweighed by the introduction of subtle errors that can be very difficult to find and fix.

Early experiments with new and disruptive parallel programming models are also underway. For example, the OmpSs programming model (Duran et al., 2011) is being used to investigate the performance of OpenIFS, but the porting involved is not straightforward, and it is clear that the full benefit of the new programming model cannot be exploited starting from the code

— the developer needs to begin with a deep knowledge of the algorithm (as opposed to it's exposition in legacy code) to fully exploit any benefits accruing from a new programming approach.

Fu et al. (2016) took another approach in a directed port of an entire model, in this case a much larger code — the Community Atmosphere Model. This work involved extensive refactoring and optimisation, albeit with a conscious effort to minimise manual refactoring (and downstream maintenance issues) and mainly relying on source-to-source translation tools. In this

case, because the target platform (the Sunway TaihuLight supercomputer) is constrained in memory performance, much of the effort was on minimising memory footprints. Without this work, the unique architecture of the TaihuLight would not have been fully utilised, with it, a speedup of a factor of two was possible. Notable in their list of key activities was the design of tools to manage the process: a loop transformation tool and a memory footprint analysis and reduction tool.

All these approaches were effectively addressing fine-grained parallelism in some way or other without addressing coarser

grained concurrency, and all involved various levels of "intrusion" into code, from adding/changing codes, to complete rewrites or translations. Such changes all risk the ongoing evolution of the codes - it is important that codes remain easy to work with so the scientific content can continue to evolve. All found that performance improvements were modest — primarily because weather and climate codes do not have "hot spots" that can be identified and optimised. Typically, the low hung fruit of such optimisation has already been harvested, and this with the complicated inter-relationships between dynamics and physics

discussed in section 2.2 coupled with the sheer number of geophysical processes involved, has led to very "flat" codes. For example, the atmosphere component of a current climate model running using 432 MPI tasks, has no subroutines that take more that 5% of runtime except for halo exchanges (of which more below). Only 20 of the 800+ subroutines involved take 1% or more of the runtime! However, despite the fact that performance improvements were modest, in all but the earliest work the lessons learned form the basis of ongoing endeavours.

The obvious solution to these problems is to attempt a separation of concerns to break the overarching modelling problem into more discrete activities and somehow address the flat computational profile within that paradigm. In most of the remainder of this section we introduce a range of current activities which span the breadth of weather and climate modelling that are aimed at some part of such a separation of concerns which we (and the institutions, projects, and partnerships we represent) are applying in the context of weather and climate modelling.

All are activities carried out in the context of wanting to address four primary aspects of modelling:

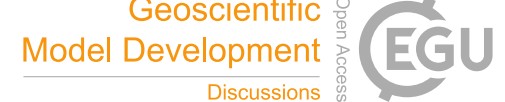

1. ensuring model quality (the models available are meeting the advancing scientific requirements),

2. delivering performance (in simulated years per wall-clock day),

3. supporting portability (so that they can run on all the relevant platforms currently in use, or likely to be in use in the foreseeable future — without excessive lead times associated with porting), and

4. productivity (the ability to work with the codes from a scientific perspective — changing algorithms, adding processes, etc, in a reasonable period of time, and without inadvertently compromising on reliability/accuracy of code).

We conclude the "recent progress" section with a brief review of how this work complements related work in other disciplines.

### 3.1  Domain Specific Languages

As discussed above, much parallelisation in weather and climate codes is delivered by individuals constructing/modifying code using libraries (such as MPI) and directives (such as OpenMP); this despite decades of research into parallelising tools. The dominant reason for this is that humans can inspect an algorithm, and with knowledge of the hardware, further exploit domain-specific knowledge to reason about how to improve performance — but the domain specific knowledge is not available to generic parallelisation tools. The result is that humans currently deliver science code that performs much better than that

produced by generic tools (although this is not the case with compilers, which generally produce vectorised assembly code which performs better than that produced by humans)

So what are the domain-specific characteristics which are being exploited by the humans? They know that weather and climate models typically employ finite difference, finite volume or finite element discretisations of the underlying equations, and as a consequence:

– Operations are performed over a mesh,

– The same operations are typically performed at each mesh point/volume/element,

– Many operations are independent of each other, allowing data parallelism,

– Operations at a mesh point/volume/elements can either be computed locally or depend on neighbouring elements (leading to nearby neighbour –*stencil* — communication patterns).

– Reductions are required for convergence and/or conservation (leading to requirements for global communication).

In addition the meshes themselves typically

– have fixed topologies,

– are structured or semi-structured (quasi-uniform),

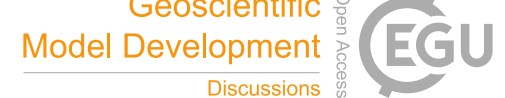



- have resolution in the vertical which is much smaller than the horizontal (leading to 2D + 1D meshes rather than full 3D meshes)

- have data structured or unstructured in the horizontal and structured in the vertical

And we can further note that

- Dynamical cores are mostly independent in the vertical, and

- Physics (parameterisations) are mostly independent in the horizontal.

These characteristics fundamentally affect how to efficiently parallelise a code. For example, one would parallelise a 3D finite element model of an engine, which would typically be unstructured in all dimensions, very differently to a finite element model of an atmosphere. In the latter case optimisations can make use of the knowledge that one direction — the vertical — is
typically more structured and has lower dimensionality.

Domain-specific languages (DSLs) provide constructs that are suitable for the domain in question, rather than being general purpose. This approach can lead to a more intuitive (higher level) interface for the scientific programmer, since they will be dealing with a much simpler, more constrained language (in comparison with general purpose languages). It can also lead to a better separation of concerns between the algorithmic description (using the DSL) and optimisation. For example, Weather
and Climate codes, in the form described above, have two main communication patterns, nearest neighbour (stencil) accesses and global reductions. Therefore, the optimisation scheme associated with such a DSL need only understand and reason about these two forms of communication pattern.

DSLs are not necessarily designed for entire communities, they can be targeted at particular requirements within communities. For example, there are two exemplars of the DSL approach currently in use by the weather and climate community:

- GridTools (Fuhrer et al., 2014a), initially for the COSMO consortium. The GridTools project includes a wide range of activities, some of which are discussed further below, but the DSL component of the work has evolved from an earlier DSL (Fuhrer et al. 2014b and Leutwyler et al. 2016).

- PSyclone, for a new UK model — LFRic. PSyclone has evolved from the Gung Ho dynamical core project (Ford et al., 2013)).

Each has been expressly developed to separate the science specification from the hardware optimisation, so that the science writer only needs to write the science code, with separate implementation and optimisation for parallelism. This allows the science to be written once, with (potentially multiple) implementations tailored to particular architectures in such as way as they do not intrude on the maintainability (and evolution) of the science code. Both groups are already exploiting this potential to separately target different architectures (e.g. see the discussion in section 3.2 below).

Both GridTools and PSyclone make use of Domain Specific Embedded Languages (DSELs), i.e. their languages are embedded within an existing language, C++ and Fortran respectively. In both cases this choice was made as it was felt that the



combination of available tools and community familiarity bought important benefits — but clearly different choices as to those benefits were made.

They also differ in their original target domains. The origin of GridTools was a DSL designed for a finite difference regular latitude-longitude local area model whereas PSyclone was designed for a finite element model with an irregular mesh. However, the GridTools formalism is being extended to support irregular finite difference models and PSyclone is being extended to support regular finite difference models.

The approach to optimisation in the two systems is also quite different. In GridTools a particular back end (optimised by an HPC expert) is chosen and the science description is mapped to that particular back end via C++ templates. In PSyclone, an HPC expert specifies a recipe of optimisations which are then used by Psyclone to process the DSL input to produce code optimised for a particular architecture.

Despite these differences, both approaches rely on HPC expertise to deliver architecture dependent performance. The DSL approach expedites the separation of concerns in such a way that computational experts can concentrate on the optimisation for given hardware, allowing different individuals to concentrate on different parts of the problem (e.g. Schulthess, 2015).

The use of DSLs has also expedited the involvement of vendors who usually have insight into how best to exploit their hardware. However, their ongoing deep engagement will be dependent on the size of the market — clearly for any given DSL, the more widely it is used, the more incentive there is for vendor engagement. Both projects recognise the importance of "growing their market" to maintain this incentive.

## 3.2 Weather and Climate Dwarfs

As already noted, many components within weather and climate models have very flat computational profiles, but one approach to dealing with this is to identify "mini-apps" (analogous to the so-called Berkely dwarfs, Asanovic et al. 2006), which are exemplars of patterns in the code (e.g. common communication behaviours), so that performance issues can be investigated (or predicted) by using a small code base, typically a fraction of the size of the larger code (Heroux et al., 2009). When combined with a DSL approach, lessons learned from investigating mini-apps could be exploited more widely within the codes with minimal human intervention.

The ESCAPE (Energy-efficient Scalable Algorithms for Weather Prediction at Exascale; www-hpc-escape.eu) project is investigating this approach for weather and climate with a specific focus on energy efficiency and emerging processor technology. ESCAPE initially defined dwarf categories which address key aspects of such models by including key components of the ECMWF spectral model IFS, its semi-implicit time stepping formulation and the semi-Lagrangian advection scheme. These allowed investigation of aspects of both global and limited-area prediction in the context of wave, ocean and land surface components.

The IFS model takes one approach to the discretization of prognostic partial differential equations and their evolution in time defined on a grid. However, alternative discretizations are also realised with a finite-volume approach, which added dwarfs related to the advection scheme (Kühnlein and Smolarkiewicz, 2017) and a 3D grid-point based elliptic solver (Smolarkiewicz et al., 2016), explored with both structured and unstructured grids. Two further dwarfs have also been chosen as representatives



| | | |
|---|---|---|
| D | Spectral Transform; SH | Spherical harmonics based transform to facilitate semi-implicit solvers on the sphere. |
| D | Spectral Transform; biFFT | A 2D Fourier spectral transform for regional model applications. |
| D | Advection; MPDATA | A flux-form advection algorithm for sign-preserving and conservative transport of prognostic variables and species. |
| I | 3D Interpolation; LAITRI | Interpolation algorithm representing a wide class of interpolation and remapping uses in NWP & Climate. |
| D | Elliptic Solver - GCR | An iterative solver for the 3D elliptic problem arising in semi-implicit time-stepping algorithms. |
| D | Advection - Semi-Lagrangian | An implementation of the semi-Lagrangian advection algorithm. |
| P | Cloud Microphysics; CloudSC | Cloud microphysics scheme from IFS, an exemplar of a range of physical parameterisations. |
| P | Radiation Scheme; ACRANEB2 | An exemplar radiation scheme used in regional NWP modelling. |

**Table 1.** The ESCAPE dwarfs — a set of mini-apps providing a sample of characteristic behaviours of weather and climate model sub-components. The "D" category dwarfs are related to the dynamics, the "I" dwarf is general purpose, and the "P" dwarfs represent physics related issues.

of the (many) physics parameterisations in models — a cloud microphysics and radiation scheme (Mengaldo, 2016; Mašek et al., 2016).

The computational performance challenges are fundamentally different between dwarfs just like their functional separation within an Earth system model. For example, spectral transforms are very memory and communication bandwidth intensive. Three-dimensional elliptic solvers are compute intensive and affected to a greater extent by communication latency. The semi-Lagrangian advection scheme is also communication intensive and depending on implementation may suffer from load-balancing issues. All three of these dwarfs have limited scalability because they involve data transfer and computations of global or regional fields - opposite to the nearest-neighbour principle. By contrast, parametrizations for cloud microphysics and radiation are very compute intensive but scalable, as they are applied to individual vertical columns that can be distributed across both threads and tasks in chunks aligned with the available memory. Hence, performance enhancement options need to be explored differently for each dwarf based on the range of the available processor types and considering that all these processes need to be present in an Earth-System model.

For each dwarf, ESCAPE targets performance on existing and emerging processor technologies (specifically Intel Xeon, Xeon Phi, and NVIDIA GPGPU and a novel technique employing optical interferometry particularly suitable for Fourier transforms), but it is also targeting programming methodologies. Of necessity this involves comparing modified and existing MPI and OpenMP implementations with new OpenACC and DSL based implementations (the latter using GridTools).

### 3.3 Data Models

The earliest weather and climate models discretised equations using a grid with one data point per cell - this led to easy data layouts, with data contiguous in memory, and relatively straightforward algorithms to find vectorisation. Modern grids and





meshes are more complicated, not always structured, and often with multiple points per cell. Finite element methods add additional complexity: instead of points representing values associated with each cell, basis functions are defined within each cell, and field "values" are established by parameters per cell known as "degrees of freedom (dofs)" which, for each basis function, establish how the field varies *within* the cell. These dofs are then associated with mesh entities such as volumes,

surfaces, edges and vertices.

There is no established convention or standard for layout in memory of these more complicated grids and meshes or how the various cell entities are associated with each cell. Current best practice is tightly coupled to knowledge of the grid being used; ECMWF's IFS model, for example, has a data structure designed specifically for the reduced Gaussian grids which arise from exploiting the spectral transforms used in the dynamical core (Hortal and Simmons, 1991). But the spectral transform

method may no longer be attractive on future architectures due to its inherent all-to-all global communication pattern, and this is recognised in the exploration of alternative approaches via the extra dwarfs discussed above. Some of these other approaches rely on nearest-neighbour communication patterns, and may even be applied to unstructured grids, leading to the requirement for different data structures to get maximum performance; potentially with different variants depending on the target hardware!

The Atlas library (Deconinck et al., 2016, 2017) is an attempt to address these issues. It provides a general framework

for representing model data structures on different grids and meshes relevant to NWP and climate simulation. It is an open-source object-oriented C++ library with a modern Fortran interface, designed to be maintainable, and non-intrusively adapt to emerging hardware technologies and programming models. This is achieved by a design with a clear separation of concerns, that for example, provides a clean interface for exploitation by a DSL and allows multiple backends for actual field data storage: a CPU backend, and a GPU backend, with the capability to synchronise with each other seamlessly.

The Atlas library provides a toolkit of functionalities, ranging from low-level Field containers to high-level mathematical operators and remapping operators. One example of the Atlas object orientated design is the way that the grid structures are designed and exploited. We have already noted in section 3.1 that weather and climate models have common properties that can be exploited, and grid type are one such common property, although the grids used vary: wth regular grids, reduced Gaussian, icosahedral, and cubed sphere grids, all in common use. As a consequence, Atlas provides a classification of grid objects,

and supports their configuration for different model variants at different resolution via simple resolution parameters. Figure 4 shows how such grid objects can be constructed as specialisations of common characteristics with resolution parameters. More specific grids may be added non-intrusively exploiting the object orientated design.

Figure 5 sketches how these *Grid* objects can be used in a workflow to configure how memory structures can be set up using Atlas. A *MeshGenerator* can, using a given *Grid*, generate a parallel distributed *Mesh*. A *Mesh* partition provides an unstruc-

tured ordering of nodes, that connected through lines (a.k.a. edges) form two-dimensional elements (triangles or quadrilaterals). Connectivity tables essentially link elements, edges, and nodes. Atlas further provides functionality to grow each *Mesh* partition with a halo so that parallelisation strategies requiring halo-exchanges can be formulated. Using a *Grid* or a *Mesh*, a *FunctionSpace* can be created, which sets up how fields are discretised and parallelised.




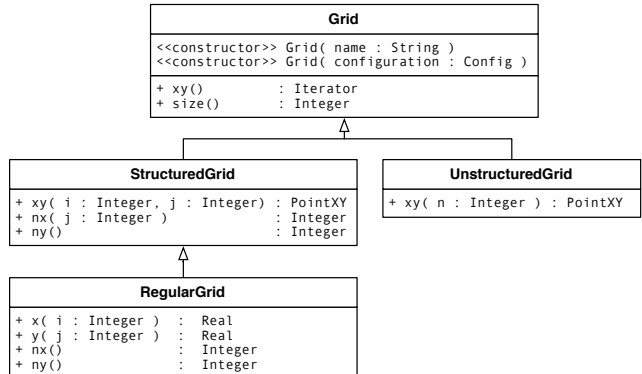

**Figure 4.** Object orientation and the *Grid* classes supported in Atlas. Specific model grids can be constructed as instances of one of these classes. For example, the current ECMWF high resolution (9km) octahedral reduced Gaussian grid is an instance of the StructuredGrid, but can be referred to by the name Ø1280 which it inherits from the Grid class.

A *NodeColumns FunctionSpace* is available to describe for every node of a horizontal mesh, a contiguous-in-memory structured vertical column. A *Spectral FunctionSpace* describes a field with spectral coefficients. Additional *FunctionSpace* types can be created to describe discretisation for continuous or discontinuous spectral element methods, (e.g. Marras et al., 2015).

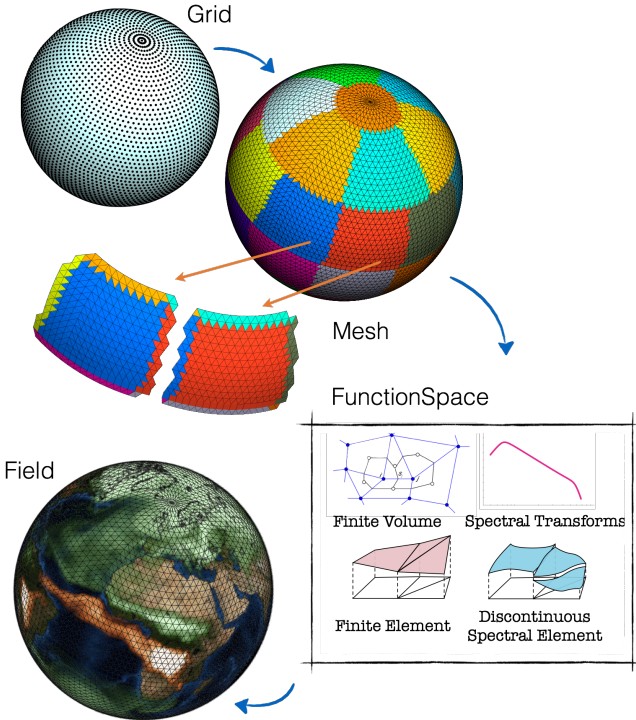

**Figure 5.** Conceptual workflow on how to use Atlas with most applications.



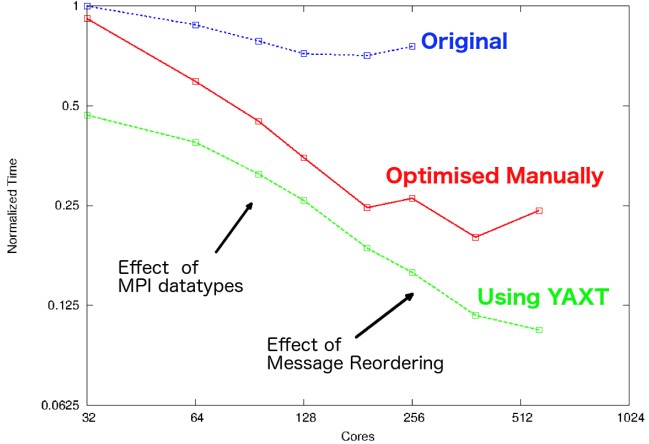

**Figure 6.** The influence of using a library to manage MPI communications for domain decomposition: experiences using YAXT to replace the existing ECHAM usage of MPI in a T63L47 model. The YAXT optimised library (green) can lead to factors of 10 improvement in the time taken to do transpositions of data over simple implementations (blue) and factors of two over manual optimisation (red, which was effectively aggregation of multiple transpositions into bigger messages). In this case the two main optimisations in YAXT were avoiding additional data copies by the usage of MPI datatypes, and reordering of messages to avoid congestion, with the latter becoming more important with higher core counts.

Within ESCAPE, Atlas is being used as the foundation for the evolution of the NWP and Climate dwarfs. As a consequence Atlas acquired improved mesh coarsening strategies, a GPU-capable *Field* data storage, and support for regional grids with projections and checkerboard domain decomposition. Clean interfaces in both Atlas and GridTools are being used to exploit Atlas memory storage to allow GridTools to compute stencil operations on unstructured meshes.

### 3.4 MPI Libraries, coupling and IO

All methods of coarse grained parallelisation which do not use global address spaces exploit the MPI paradigm — whether it is managing domain decomposition, coupling components, or exploiting I/O servers.

Exploiting MPI directly is difficult, and models typically exploit some sort of middleware library such as Generalised COMmunications (GCOM) used by the Met Office UM or YAXT (Yet Another eXchange Tool, https://doc.redmine.dkrz.de/yaxt/

html/da/d3c/concept.html) under testing for use in ICON. Such tools can simplify the usage of MPI and therefore make the performance potential of data layout and data independence accessible (e.g. see figure 6).

The complexity of data transpositions and MPI communications increase in the context of coupling, and these too need to be understood in terms of how best to obtain performance. Many groups are investing in coupling tools, such as the OASIS coupler widely exploited in Europe. Recognising that there is much to be learnt from these various activities, the community

has held a series of workshops on coupling, which led to the identification of a range of parameters which influence coupling and can be used in a benchmarking strategy (table 2).





| Grid TYPE of the coupled components |
| Grid SIZE of the coupled components |
| Number of processes for each component |
| Layout of the components on the available cores |
| Number of fields exchanged between the components |
| Disparity in the grids between components |
| Scheduling of the components and coupling |

**Table 2.** A set of characteristics that influence coupler performance.

With these characteristics in mind, the IS-ENES2 project established a programme of investigation into coupling performance by developing some standalone components suitable for coupling tests, and carried out some initial investigations testing the impact of the number of cores per component, the grid sizes, the grid aspect ratios and orientations, and having different numbers of cores for different components.

As a proof of concept, these coupled test cases were run using five different coupling technologies in different configurations on three different platforms: Bullx at CINES in France, Cray XC40 at the UK MetOffice, and the Broadwell partition of Marconi at CINECA in Italy. Thus far the work has demonstrated the utility and versatility of the benchmarking environment; future work will focus on learning lessons from comparisons of performance (Valcke et al., 2017).

### 3.5 Tools to support porting, optimisation and performance portability

Automatic code parsing tools provide a route to code migration (e.g. Fu et al., 2016, as discussed above), but their efficiency depends on the internal models of how codes are, and can be, organised, and in particular the presence or absence of pre-existing manual optimisations, which if not understood by the tool, can require significant manual effort to remove (before and/or after the use of the tool).

It is clear that achieving performance requires a comprehensive understanding of what existing model components/dwarfs

are doing, and performance-portability requires designing code that will still have performance on alternative architectures. With the complexity of modern codes, this can no longer be efficiently done with access to simple timing information, more complicated information needs to be gathered and presented for analysis by developers. Such information is normally provided by performance tools.

Performance tools can provide the support and the freedom to the developer to view all the different levels of concur-

rency (from the hardware counters underpinning fine-grained concurrency to the load-balancing involved in coarse-grained concurrency). While it is possible to collect a vast amount of information by re-linking entire codes to performance monitoring libraries, often results are enhanced by manually identifying hot-spots and extracting additional trace information, and automatically generated call-trees can be enhanced by additional instrumentation.





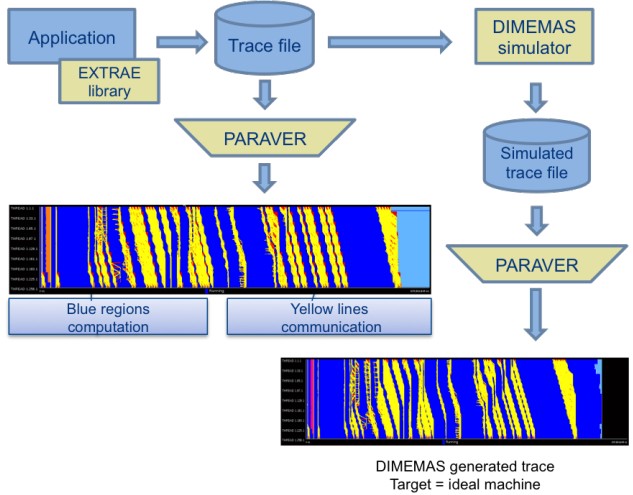

**Figure 7.** Understanding model performance on existing machine generally requires instrumenting the model code to output "trace" information that can be analysed and visualised by a dedicated application. This information can also be used, when combined with machine information, to predict performance (issues) on alternative architectures. This figure depicts the use of PARAVER (Labarta et al., 2005) for collecting trace information, and DIMEMAS (Gonzalez et al., 2011) for simulating performance on alternative architectures.

In particular, like automatic code migration tools, the tools which generate trace files, such as the EXTRAE library underpinning the generation of trace files for PARAVER or DIMEMAS, depend on internal understanding of the programming models in use in the target codes. It is this understanding that allows them to automatically trap library calls and read hardware counters without code invasion, but it is precisely because of this dependence on generalised programming models, that extra
manual intervention can be necessary to capture domain dependent information in trace files.

The resulting trace files can be analysed directly to help improve performance on any given architecture, and also used to predict performance on alternative architectures (see figure 7). Any extra levels of invasive alteration of code add further levels of programming complexity, and even with intuitive analysis tools (e.g. see figure 8), considerable time and expertise is require to interpret the outputs. However, the regular use of such tools by DSL developers and those evaluating dwarfs is one route to
10 maximising the benefit of these tools without all the community needing to gather the relevant expertise.

### 3.6 Related Work

Weather and climate computing is not alone in facing these problems. For example, the notion of DSLs as a solution has a tried and tested heritage — examples include the Kokkos array library (Edwards et al., 2012), which like GridTools, uses C++ templates to provide an interface to distributed data which can support multiple hardware backends, and from computational
chemistry, sophisticated codes (Valiev et al., 2010) built on top of a toolkit (Nieplocha et al., 2006) which facilitates shared memory programming. Arguably DSLs are starting to be more prevalent because of the advent of better tooling for their development, and because the code they generate can be better optimised by autotuning (Gropp and Snir, 2013). However, in





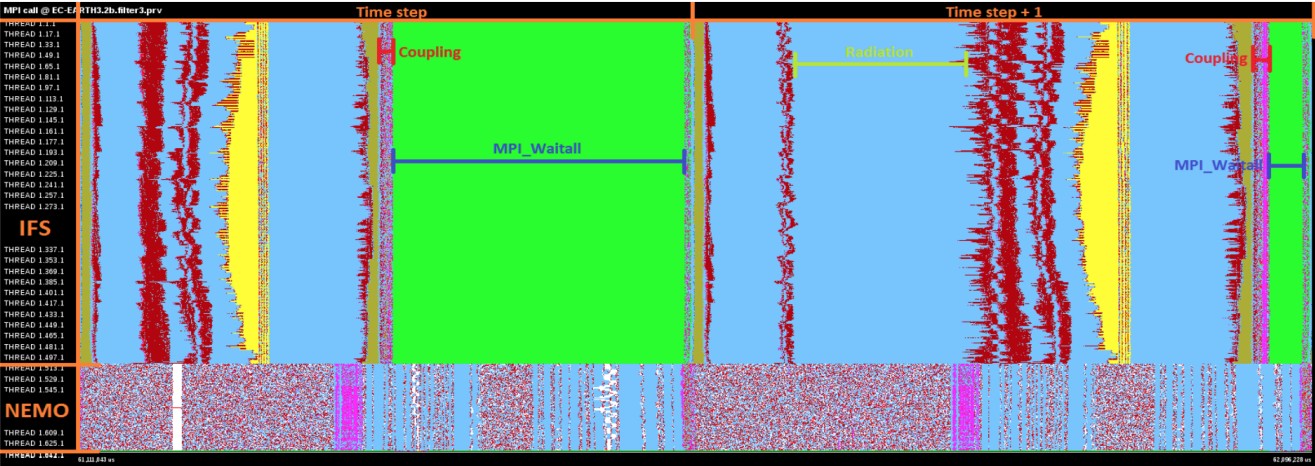

**Figure 8.** Example trace information generated with EXTRAE used to analyse high resolution coupling in EC-Earth. This view shows MPI calls during the execution (one color per call). On top, 512 MPI tasks for IFS atmospheric model and bottom, 128 MPI tasks for NEMO ocean model. This fine-grained performance information yields information as to how to get better performance out of coarse grained concurrency.

our case, we still believe that human expertise has a primary role to play, but this expertise can be better deployed by supporting DSLs than entire code-bases.

The mini-apps or dwarfs approach we describe above is also starting to become more widely prevalent, mainly because it provides a tractable route to both predict performance and develop and test performance optimisations for new hardware.
Its applicability to weather and climate modelling follows directly from the fact that the source of the methodology was in disciplines where similar flat multiphysics profiles were involved. The concept of attempting to represent a cross section of applications or application characteristics is becoming well recognised (e.g. Martineau et al., 2016), and others are investigating applicability to environmental models (e.g Stone et al., 2012), and noting the importance of representativeness in the selection of the mini-apps.

Problems of establishing appropriate data structure, generating meshes, and simplifying the related interfaces and workflows are prevalent across all communities addressing the solutions of partial differential equations, whether on a sphere or not, with decades of relevant research for deployment on parallel computers. More recently, work on data structures has begun to address cache awareness, the use of space filling curves (e.g. Günther et al., 2006), and a wide range of mesh generators are available. The issue for weather and climate modelling is not so much what relevant work there is, but how best to establish domain
specific tooling which yields scientific and software development productivity across the community.

Further up the software stack, model couplers, frameworks, and I/O servers are all tools where the domain specificity dominates, albeit many communities have analogous problems. Tools for evaluating code performance are common across communities, with important related work in the establishment of standards for trace file formats (e.g. the Open Trace Format Knüpfer et al., 2006) and common notations for adding directives for extra performance related information.



## 4 Discussion

We have introduced a range of modes of hardware and software complexity, requirements for concurrency, and recent progress in addressing the evolution and performance of weather and climate codes. We have done this in the context of wanting to ensure model quality, keep/improve performance, sustain portability (in the face of the Cambrian explosion in computing), and

maintain productivity — the continual increase in the scientific capability of the models.

In our discussion we address these requirements and their dependency on current trends on complexity. It will be seen that we contend that progress cannot be maintained on the existing trajectory, and so in the final part of the discussion we outline our manifesto as to what the community needs to do.

### 4.1 Requirements

Existing models are known to have high levels of software quality (Pipitone and Easterbrook, 2012), and maintaining quality models in the face of increasing scientific demands on what is simulated, ensemble size, and resolution etc, will be crucial. It is these that define scientific quality — a model could be very high performance and portable, but not be suitable for the scientific objectives, but there is a spectrum between "not suitable" and "ideal". In fact, some level of model quality is often compromised for performance, for example, resolution and complexity are routinely sacrificed to fit weather forecasts in a

particular time window or to reach a requisite number of simulated years per day for climate.

It is not so well known that quality is also often compromised for productivity and portability. One of the reasons for the popularity of the Weather Research and Forecasting model (WRF, Skamarock et al., 2005) is that it is easy to modify and run, and this productivity/portability benefit outweighs the use of alternative models even when they are perceived to deliver better quality weather simulations.

Until recently, given the dominance of X86 systems, performance has been relatively independent of platform; once compiler differences have been negotiated, getting the best performance has been mostly about understanding how to exploit processor decomposition given the interconnect, and ignored by much of the modelling community, except by those with the most demanding applications (typically in weather predictions in tight wall-clock time windows, or in long, complex or high-resolution climate runs). Nonetheless, performance also involves the need to optimally fill machines (or execute large ensembles), and

so optimising performance can also mean addressing throughput rather than speed (Balaji et al., 2017) — meaning that performance optimisation should not only be about good fast code, but about evaluating speed for a variety of situations within a large potential parameter space (e.g processor decomposition and placement).

Portability, that is the ability to migrate codes between architectures and achieve acceptable performance (with confidence that the code is delivering the same scientific results, e.g. Baker et al. 2015) is very important, whether to ensure codes can

be executed on many platforms by a wide community, or to ensure that the codes will work on future architectures. In many groups it is seen as one of the most important considerations (e.g. Norman et al., 2017). However, as we have seen, obtaining ideal performance on multiple architectures with the same code is problematic, and so performance portability is always a



compromise. In fact, it has been said that "the real goal is probably better labeled maintainability than strict portability" (Norman et al., 2017, again).

The difficulty we now face is that fact because of the oncoming heterogeneity, even vendors are suggesting that the community have to face a world in which one can only have two of performance, portability, and productivity — a happy compromise is no longer possible. While this is to some extent hyperbole, it is certainly true that much more effort needs to be devoted to achieve these simultaneously, and that quite major changes in approach (e.g. the use of DSLs) are needed — so it is the amount of effort needed, the change in approaches required, and the timescales in play, that lead to the "cliff/edge chasm" motif for this paper.

## 4.2 Making Progress

To make progress in the science, existing weather models are being extended to seasonal and decadal scales by running with additional components (ocean, sea and land ice), and existing climate models are confronting the need to start from prescribed initial conditions and run at increased resolution. New models are being developed, but the existing times scales from development to production science serving large communities is currently of order a decade. All these models need to exploit all the avenues of concurrency outlined in section 2.3, and, at the same time support multiple architectures. How then, can we make progress?

There are essentially four strategies to sustaining a modelling programme:

1. Develop a model and all the supporting components from scratch,

2. Build a particular piece or pieces of a model, and utilise components and/or infrastructure from elsewhere,

3. Utilise someone else's model, but tinker with some aspects of it, or

4. Simply configure and run models developed elsewhere.

Which strategy is most appropriate depends primarily on the resources and skillsets available, and the productivity possible. Only a handful of groups can afford the first strategy, and the further down the list one goes, the less control one has over the future trajectory that is possible for that modelling programme because of the external dependencies.

Risks range from not being able to move fast enough to meet science goals, to having key capability removed because the external group has chosen either to do something completely different, or change key aspects of the interfaces upon which the model depends. In the worst case, having outsourced some capability, when that is removed, the internal group may no longer have the capability to replace that capacity. Mainly for these reasons, large (primarily) national modelling endeavours prefer to keep as much development as possible in house. However, looking forward, it is not obvious that even large national groups have the internal resources to *both* keep up a production line of incremental model improvements associated with meeting near-term scientific (and/or operational) requirements *and* identify and taking the requisite steps necessary to develop codes which can hit quality + performance + portability + productivity requirements using next generation computing — particularly when

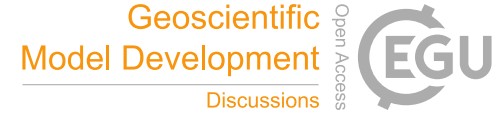



the latter is unlikely to be achievable with small steps. This *is* recognised in some large institutions, for example, the Met Office is investing in OASIS, NEMO and XIOS, all of which have or will replace internal developments.

If the existing incremental development method will not provide more than modest improvements in performance at very low levels of computational efficiency, how then can progress be made? We would assert that the way forward has to involve
the community making better use of the collective resources, and more groups utilising components, particularly infrastructure components, developed elsewhere. It is such shared libraries and tools that we see as the "spans" that will need to be utilised to cross the "chasm" of model development ahead. That said, we fully recognise the importance of competitive evolution: the optimal number of libraries and tools is not one of each, two or three might be the right answer, but it should certainly not approach $N$, where $N$ is the number of large modelling groups.

We have introduced a range of such tools and libraries, and methodologies for making progress such as the identification of dwarfs, the publication of data structures, the elements necessary to establish a comparison of coupling technologies, and to evaluate performance. Within some of that work there is a clear aim to develop and specify interfaces that can be re-used in different ways (for example, the interfaces which allow GridTools to interface directly with the Atlas data structures), but in the main many of these (and similar) projects have still been developed by small groups who have not had the time, mandate,
or wherewithal to publicly promulgate requests for requirements or specifications, and then build to documented interfaces.

Progressing at the community level will require improved methods to allow the community to discuss, specify, design, develop, maintain, and document the necessary libraries and tools. The weather and climate community does not have a great track record at sharing such tools, although in recent years necessity has begun to influence practice — with tools such as OASIS becoming more prominent in more places. One of the reasons for this lack of sharing is in part the lack of a commonly deployed
structured approach to sharing, one that maximises delivery of requirements, while minimising risk of future technical burden — the sort of approach that has delivered the MPI libraries upon which nearly all of high performance computing depends. While a fully fledged standards track is probably beyond the will of the community at this point, it is certainly possible for the community to investigate, and then engage in, more collaborative working.

Such steps would begin with institutions and modelling groups recognising the scale of the problem ahead, and recognising
that:

- business as usual, consisting of modest incremental steps, is unlikely to deliver the requisite next generation models,

- they do not have enough internal resource to take the leap to the next generation alone, and most importantly,

- there are library or tool projects which they can exploit and where they can collaborate rather than compete, some of which may be from outside the traditional communities of collaborators.

Institutions which are most likely to be able to seize these collaborative opportunities are most likely to share the following characteristics: They will

- Have understood the issue fully at the management level, the science level, and in the infrastructure teams,

- Be able to reward individuals for innovation in, and/or contributions to, external projects,

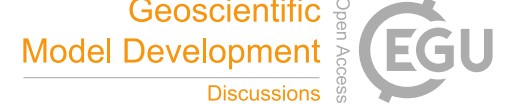



- Recognise the benefit of external scrutiny and avoid situations where they are for ever arguing that their code is "not ready to share',

- Have the courage to stop existing activities and pickup and use/integrate third party libraries and tools, and

- Have the ability to recognise the cost-benefit trade-off between "doing it themselves" and contributing intellectually and financially to third party solutions, and

- Be ready to apply more sophisticated and complex software engineering techniques, and encourage more computational science research.

Projects which would be suitable for collaborative working will share some common characteristics. They will:

- be open source *and* have an open development process,

- have clear goals, scope, and where appropriate, deliver stable software interfaces,

- have a mechanism to understand and respond to the timescales of collaborators (that is, some sort of governance mechanism which assimilates and responds to requirements),

- potentially be able to accumulate and spend funds to provide user-support, training, and documentation,

- be not initially disruptive of existing solutions, and ideally

- engage both the scientific community and vendors (compare with MPI where vendor implementations are often key to enhanced MPI performance).

## 5 Conclusions

Having recognised the scale of the problem, the characteristics of institutions which can work collaboratively, and the nature of good collaborative projects, the question then becomes: what are the important tools, components, or dwarfs that will be needed to bridge the metaphorical chasm between current practice and what is necessary to sustain weather and climate modelling?

Clear scope is crucial, from tools to software aiming at particular parts of the problem. We have already seen a list of types of concurrency categorised into fine-grained and coarse-grained, and a list of "weather and climate" dwarves being investigated within one project — but a more sophisticated taxonomy will be required to allow the community to better develop complementary shared tools and libraries.

Such a taxonomy will cover at least:

- Tools for exposing mathematical algorithms for implementation on a sphere (domain specific languages),

- Tools for describing and using data models for the variables in those algorithms (including stencils for computation),

- Mathematical Libraries such as Fast Fourier and Spherical Transforms,

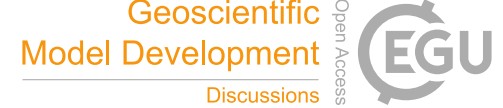



- Solvers which can exploit specific data models and algorithms,

- Interpolation and Regridding Libraries,

- Embedded and standalone visualisation tools,

- Exchange libraries (for problems ranging from domain halo exchanges to 3D field exchange between high level components),

- Fully fledged couplers (e.g. OASIS) and frameworks (e.g. ESMF),

- I/O servers (such as XIOS),

- Data Assimilation tools such as minimisers and adjoint compilers,

- Clock, calendar, time-stepping and event handling libraries (events such as "do at first time step, do every three hours, etc),

- Testing Frameworks,

- Performance and Debugging tools,

- Domain specific tools for automatic code documentation.

Many of these elements have been subject to classification, whether as part of specific projects (e.g. ESMF, or GridTools – figure 9), or in more theoretical approaches such as Rugaber et al. (2011), but a more comprehensive and complete approach is needed before it will be possible for groups to commit to building for, sharing with, and depending on, each other. In particular, the interfaces between the various elements need attention so that a more a la carte approach can be established as elements progress through phases of maturity and/or usability. Such interface definitions would eventually include strong software contracts, allowing the construction of complex software systems that would reliably meet requirements and expectations.

The ESCAPE approach provides a generic template for both bottom-up and top-down approaches to the problem. The top-down direction is driven by the needs of NWP and climate and subsequent selections of critical sub-components used in time-critical global, limited-area, coupled or uncoupled Earth-System simulations. The bottom-up direction lays the foundation for exploiting different types of processor technologies by developing specific and scalable (numerical methods) research support libraries, and performs code adaptation to address computing bottlenecks that are dwarf specific. However, this project driven and historically short term form of collaboration, is unlikely *on its own*, to provide the stable foundations necessary to establish the necessary community wide element/interface consensus, not least because of insufficient guarantees of long-term maintenance and evolution.

Given the existing lack of consensus around approaches and key interfaces, it may be that progress still needs to be based around such ad-hoc consortia, but within that we believe it is crucial that projects and consortia strive towards the open requirements gathering and open development processes outline in section 4.2 and attempt to go beyond just solving their own





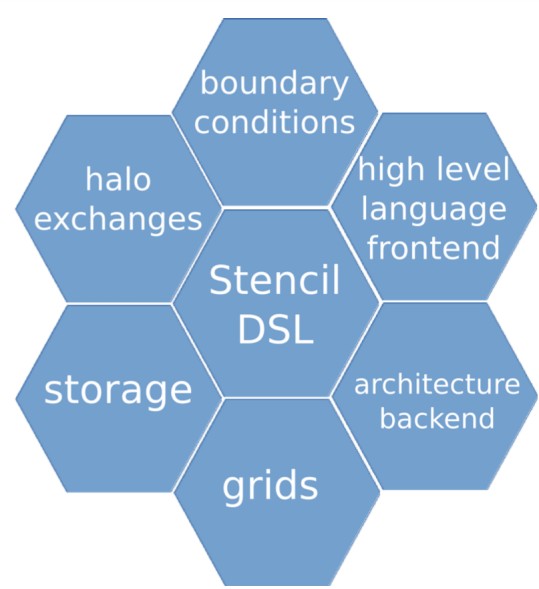

**Figure 9.** The GridTools separation of concerns provides some elements of the taxonomy of components needed by modelling programmes as well as identifying some of the interfaces needed.

problems. This will require projects to commit more effort towards outward engagement, before, during, and after development as well as to more extensive documentation and training — but the benefit to institutions and funders will outweigh the initial costs, not least because the more standardised and widespread weather and climate tooling becomes, the more incentive there will be for industry to invest in contributing to solutions.

5     Getting projects established which do this will require managers, scientists and software engineers to all recognise the scale of the problem and the need to go beyond the short-term aims of their institutions. This will be especially problematic since most of the relevant institutions are focused on delivering advances in environmental science (individually or collaboratively), in most cases orthogonal activities aimed at collaborative advances in the underlying software infrastructure are harder to achieve!

10  **6  Code Availability**

This paper describes a range of models, tools and projects. There are three models discussed, further description and information as to how to obtain code and appropriate licenses can be found via the following URLS: http://www.metoffice.gov.uk/research/modelling-systems/unified-model, https://software.ecmwf.int/wiki/display/OIFS/OpenIFS+Home-systems/unified-model, and https://www.gfdl.noaa.gov/hiram/.





*Author contributions.* All authors contributed content, either directly, or via significant contributions to initiating and organising the "Crossing the Chasm" meeting held under the auspices of the IS-ENES2 project in Reading, UK, in October 2016.

*Competing interests.* None

*Acknowledgements.* This paper reports European research funded by the following FW7 and H2020 research and innovation projects: IS-
5   ENES2, under grant agreement 312979; ESCAPE under grant agreement No 671627; and ESIWACE, under grant agreement 675191. The
authors acknowledge useful conversations with, and input from, V. Balaji, Terry Davies, Peter Fox, Rich Loft, Nigel Wood, and Andy Brown
and the input of other participants at the "Crossing the Chasm" meeting, in particular Jean-Claude Andre, Joachim Biercamp, Antonio Cofiño,
Marie-Alice Foujols, Sylvie Joussaume, Grenville Lister, Alastair Mckinstry, Annette Osprey, Øyvind Seland, and Manuel Vega.



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
