# Peer review of "Crossing the Chasm: How to develop weather and climate models for next generation computers?"

_Geoscientific Model Development, 2017_

## Referee Comment (RC1) · Anonymous Referee #1 · 6 Oct 2017

This paper provide a good (referenced) summary of the pressures that are growing the complexity of Earth system models and consequently limiting our ability to exploit emerging high-performance computing platforms. Most of these pressures should be familiar to practitioners and the topics have been discussed in varying detail in other articles.

However, some of the proposed approaches for solving these issues will be less familiar to many in the community: domain specific languages, "dwarfs", data models, etc. In this respect, the paper is an excellent review article that provides ample references for researchers to become familiar with these potential solutions.

[Figure]

Ultimately, though, what the authors are proposing is a large community-wide coordinated effort to (further) distribute the burdens of software development. By reducing duplication and emphasizing separation-of-concerns, the hope is that models can leverage a more powerful suite of infrastructure tools, libraries, and frameworks that can more readily address performance challenges. The challenge then becomes one of coordination within the community in the face of daunting complexity.

Some specific comments and concerns:

The discussion of DSL's could be extended by including KPP/KPPA as examples. These are tools used my atmospheric chemists to translate reaction mechanisms into executable code. The tools can generate optimized code for various architectures and are generally treated as black boxes by the end-user. While most Earth system modelers have probably not directly used these, they are likely a more familiar example and could better serve as an introduction to DSL's.

The statements on page 7 regarding PGAS languages and mixed-mode programming are perhaps a bit too strong. I can only speak for Co-Array Fortran (CAF), but there are not any obvious mechanism in that language that correspond to mixed-mode support. CAF images generally correspond to MPI processes and can be implemented to run on threads (just as MPI can) or can leverage threading under-the-hood (just as MPI can). The main point of PGAS languages, at least to my understanding, is that they provide a simpler/clearer mechanism for expressing locality than say, MPI. But they do not have a tiered approach in this respect. Things are either local (in the image) or remote (off the image).

I found that the beginning section 2.1 on Hardware Complexity to be a bit of exaggeration on the whole. Too many quoted words and exclamation points. If there is another revision, I would hope that the general tenor could be altered to match the remainder of the paper.

---

## Short Comment (SC1) · 2 Nov 2017

The ACRANEB2 dwarf listed in table 1 is restricted to longwave computations, described by Geleyn at al. (2017), doi:10.1002/qj.3006. Citation of Mašek at al. (2016) on page 18, line 1 deals with shortwave computations and should be replaced by the above.

---

## Referee Comment (RC2) · Anonymous Referee #2 · 9 Feb 2018

This paper does a good job of describing the challenges associated with development of software for climate and earth system models. While these challenges are well known within the climate community they are not always well known even in the very related but different numerical weather prediction (NWP) community.

The paper uses hyperbole in the first several sections of the paper to described several issues. In particular they use the term "an impending cliff" to describe the need to modernize climate applications to take advantage of upcoming architectures. Honestly this is overstated as the current challenges are no more ominous then the challenges that were faced in the conversion from long vector machines to scalar machines in the

late 80's and early 90's. Then as today codes evolved or became obsolete. There is no mentioned that I can see that the climate community has been here before and scientific progress was not devastated.

A similar excessive hyperbole is seen in the Hardware Complexity section 2.1 Yes, there are a number of different architectures that exist today due to greater specialization. However only a limited number are actually viable for climate and NWP. The greater diversity in architectures should really be thought of as a boon for certain classes of computing problems that were not well served by existing architectures instead of a burden to the climate and NWP community.

I found that Figure 3 does not really add much to the software complexity discussion. What they really need to emphasize in this section is not that the IFS model has a bunch of blue boxes connected together in a serial fashion, but rather that the correct data-structures and computational patterns of which different models are composed are quite varied. There is some discussion of climate dwarfs in Section 3.2, and brief mentions that IFS is a spectral transform model while HiRAM is a finite volume however. But I think this concept of different computational patterns and its impact on complexity should really be fleshed out better in the software complexity section.

One issue that I feel is missing in the discussion in Section 4 is how to manage support funding across different countries outside the European Union. For example, if I am using a I/O library developed in Germany in my US climate model, how would I get the German developers to fix a critical bug that only occurs in my US model? Or is the sharing of common software infrastructure only viable across institutions within a common economic market? I think this question becomes very critical with respect to the use of a Domain Specific Language. If the shared software component is simply a library then it is relatively easy to substitute in an alternative. However the commitment to using a Domain Specific Language is much larger and bugs that were not addressed in a timely fashion could have catastrophic impact on scientific productivity. It is for this reason that I don't see a widely adopted domain specific language, other than Fortran

of course, as a viable option.

---

## Author Comment (AC1)

**Referee #1**

We are pleased the referee found this to be an excellent review article, and noted the following points:

1. The discussion of DSL's could be extended by including KPP/KPPA as examples. These are tools used by atmospheric chemists to translate reaction mechanisms into executable code. The tools can generate optimized code for various architectures and are generally treated as black boxes by the end-user. While most Earth system modelers have probably not directly used these, they are likely a more familiar example and could better serve as an introduction to DSL's.

   - *Response:* Thank you for pointing this out, we had overlooked this example. The introduction to domain specific languages now begins with reference to an example of the use of KPP in atmospheric chemistry.

2. The statements on page 7 regarding PGAS languages and mixed-mode programming are perhaps a bit too strong. I can only speak for Co-Array Fortran (CAF), but there are not any obvious mechanism in that language that correspond to mixed-mode support. CAF images generally correspond to MPI processes and can be implemented to run on threads (just as MPI can) or can leverage threading under-the-hood (just as MPI can). The main point of PGAS languages, at least to my understanding, is that they provide a simpler/clearer mechanism for expressing locality than say, MPI. But they do not have a tiered approach in this respect. Things are either local (in the image) or remote (off the image).

   - *Response:* Thank you for pointing this out, we had over-used the phrase "mixed-mode". We believe the following text is more accurate and clear: The relevant piece of text now reads:

     > Other programming models, such as the Partitioned Global Address Space (PGAS), try to address the problem of locality of memory by a single abstraction to address both shared and distributed memory. Implementations include the C extension, Unified Parallel C, Co-Array Fortran, Chapel, and the (Open)SHMEM library.

3. I found that the beginning section 2.1 on Hardware Complexity to be a bit of exaggeration on the whole. Too many quoted words and exclamation points. If there is another revision, I would hope that the general tenor could be altered to match the remainder of the paper.

   - *Response:* Given that both referees have agreed on this point, we have attempted to moderate some of the language. In particular we have removed some of the material in section 2.1 which talks about "explosions" and "proliferation" and rowed back on exclamation marks. We believe 2.1 is now consistent with the rest of the paper.

**Referee #2**

We are pleased the referee found that we had done a good job of describing the challenges associated with software development. However, we note the following points of criticism:

1. The paper uses hyperbole in the first several sections of the paper to described several issues. In particular they use the term "an impending cliff" to describe the need to modernise climate applications to take advantage of upcoming architectures. Honestly this is overstated as the current challenges are no more ominous then the challenges that were faced in the conversion from long vector machines to scalar machines in the late 80's and early 90's. Then as today codes evolved or became obsolete. There is no mentioned that I can see that the climate community has been here before and scientific progress was not devastated.

    - ***Response:*** While we agree that this is not the first time there have been major transitions in computing, some of us were there last time, and we are more concerned this time insofar as our codes are vastly bigger, have multiple coupled components, and there is more simultaneous pressure on science productivity - the time to respond and the effort for agility is simply not there *from within existing resources*. Further, this time there is as much a focus on solution algorithms (ie changing the model) as there is on restructuring the existing algorithms. That said, we appreciate that the language could be toned down, and we have done so, while attempting to maintain a sense of scale and urgency - which we believe is simply not understood in much of our target readership. We have also added an explicit mention of the subject of having been here before (although it was implicit in the discussion at the beginning of section 3).

2. A similar excessive hyperbole is seen in the Hardware Complexity section 2.1 Yes, there are a number of different architectures that exist today due to greater specialization. However only a limited number are actually viable for climate and NWP. The greater diversity in architectures should really be thought of as a boon for certain classes of computing problems that were not well served by existing architectures instead of a burden to the climate and NWP community.

    - ***Response:*** We have moderated the language as discussed in the response to referee 1, although we do not agree that hardware proliferation within the number viable for climate and NWP will not be a problem. In our experience (and expectations of hardware over the next few years) it already is - clearly there are codes that are maintaining versions for multiple platforms, but that is not possible for all institutions/codes given the human resources we have available. For example, even in the evolution from Intel SandyBridge to Skylake, some codes can, and some codes cannot, take advantage of the changing vector units. It is the proliferation on chip that is going to be as much of a problem as the proliferation of chip types.

3. I found that Figure 3 does not really add much to the software complexity discussion. What they really need to emphasize in this section is not that the IFS model has a bunch of blue boxes connected together in a serial fashion, but rather that the correct data-structures and computational patterns of which different models are composed are quite varied. There is some discussion of climate dwarfs in Section 3.2, and brief mentions that IFS is a spectral transform model while HiRAM is a finite volume however. But I think this concept of different computational patterns and its impact on complexity

should really be fleshed out better in the software complexity section.

- *Response:* We do not entirely agree with the Referee here. We do not think it is a matter of choosing not to discuss the model sequencing in favour of discussing data structures and patterns, rather that both are important. In our experience there has been considerable interest in the sequencing of activities within the models, and that the details in that sequencing will have significant impact on the possibilities for concurrent parallelism. However, we appreciate the point that we can do more to identify the differences in, and importances of, data structures and computational patterns. As the reviewer notes, we do have material on that throughough the paper, but we have modified the material in this section to make the importance clearer.

4. One issue that I feel is missing in the discussion in Section 4 is how to manage support funding across different countries outside the European Union. For example, if I am using a I/O library developed in Germany in my US climate model, how would I get the German developers to fix a critical bug that only occurs in my US model? Or is the sharing of common software infrastructure only viable across institutions within a common economic market? I think this question becomes very critical with respect to the use of a Domain Specific Language. If the shared software component is simply a library then it is relatively easy to substitute in an alternative. However the commitment to using a Domain Specific Language is much larger and bugs that were not addressed in a timely fashion could have catastrophic impact on scientific productivity. It is for this reason that I don't see a widely adopted domain specific language, other than Fortran of course, as a viable option.

- *Response:* Beyond enjoying the (probably correct) assertion that Fortran is a DSL, we agree that the issue of reliability and responsiveness in the toolchain is crucial to uptake. In the existing text we had alluded to this at the end of section 3.1 in talking about wanting vendor engagement, and in the final list of the characteristics of successful collaborative projects, but it was not explicit enough. We have added a new paragraph on this point to conclude section 4, taking advantage of the referees example, which we think is helpful. In that text we give the example of both OASIS and ESMF, based in Europe and the U.S., both of which support software across different markets.

**Comment 1**

1. The ACRANEB2 dwarf listed in table 1 is restricted to longwave computations, described by Geleyn at al. (2017), doi:10.1002/qj.3006. Citation of Mašek at al. (2016) on page 18, line 1 deals with shortwave computations and should be replaced by the above.
   - Done.